# TGFβ and Hippo Signaling Pathways Coordinate to Promote Acinar to Ductal Metaplasia in Human Pancreas

**DOI:** 10.3390/cells13020186

**Published:** 2024-01-18

**Authors:** Michael Nipper, Yi Xu, Jun Liu, Xue Yin, Zhijie Liu, Zhengqing Ye, Jianmin Zhang, Yidong Chen, Pei Wang

**Affiliations:** 1Department of Cell Systems and Anatomy, University of Texas Health Science Center at San Antonio, San Antonio, TX 78229, USA; nipper@livemail.uthscsa.edu (M.N.); xuy4@uthscsa.edu (Y.X.); liuj8@uthscsa.edu (J.L.); yinxue0303@gmail.com (X.Y.); 2Department of Molecular Medicine, University of Texas Health Science Center at San Antonio, San Antonio, TX 78229, USA; liuz7@uthscsa.edu; 3Greehey Children’s Cancer Research Institute, University of Texas Health Science Center at San Antonio, San Antonio, TX 78229, USA; yez@uthscsa.edu (Z.Y.); cheny8@uthscsa.edu (Y.C.); 4Department of Population Health Sciences, University of Texas Health Science Center at San Antonio, San Antonio, TX 78229, USA; 5Department of Cancer Genetics and Genomics, Roswell Park Comprehensive Cancer Center, Buffalo, NY 14263, USA; jianmin.zhang@roswellpark.org

**Keywords:** acinar-to-ductal metaplasia, ATAC-seq, ChIP-Seq, Hippo pathway, TGFβ signaling

## Abstract

Background & Aims: Acinar-to-ductal metaplasia (ADM) serves as a precursor event in the development of pancreatic ductal adenocarcinoma (PDAC) upon constitutive environmental and genetical stress. While the role of ADM in PDAC progression has been established, the molecular mechanisms underlying human ADM remain elusive. We previously demonstrated the induction of ADM in human acinar cells through the transforming growth factor beta (TGFβ) signaling pathway. We aim to investigate the interaction between TGFβ and Hippo pathways in mediating ADM. Methods: RNA-sequencing was conducted on sorted normal primary human acinar, ductal, and AD (acinar cells that have undergone ADM) cells. ATAC-seq analysis was utilized to reveal the chromatin accessibility in these three cell types. ChIP-Seq of YAP1, SMAD4, and H3K27ac was performed to identify the gene targets of YAP1 and SMAD4. The role of YAP1/TAZ in ADM-driven cell proliferation, as well as in oncogenic KRAS driven proliferation, was assessed using sphere formation assay. Results: AD cells have a unique transcription profile, with upregulated genes in open chromatin states in acinar cells. YAP1 and SMAD4 co-occupy the loci of ADM-related genes, including *PROM1*, *HES1*, and *MMP7*, co-regulating biological functions such as cell adhesion, cell migration, and inflammation. Overexpression of YAP1/TAZ promoted acinar cell proliferation but still required the TGFβ pathway. YAP1/TAZ were also crucial for TGFβ-induced sphere formation and were necessary for KRAS-induced proliferation. Conclusions: Our study reveals the intricate transition between acinar and AD states in human pancreatic tissues. It unveils the complex interaction between the Hippo and TGF-β pathways during ADM, highlighting the pivotal role of YAP1/TAZ and SMAD4 in PDAC initiation.

## 1. Introduction

Pancreatic ductal adenocarcinoma (PDAC) is a deadly cancer in need of better diagnosis, treatment, and early intervention [1]. A better understanding of PDAC initiation and progression will offer opportunities for cancer prevention and for the development of new diagnostic methods and therapeutic treatments. Acinar-to-ductal metaplasia (ADM) has been observed during pancreatitis and suggested to play an important role in KRAS-driven PDAC initiation in animal models [2,3]. However, the precise molecular mechanisms of ADM are not fully understood, partly due to challenges in isolating acinar cells that are undergoing ADM. This challenge is even greater in human ADM tissue. PDAC organoid models better recapitulate human pancreatic cancer characteristics than animal models do, but they are based on tumor cells and far removed from ADM [4]. We previously identified a unique set of cell surface markers that could distinguish and isolate primary human acinar, ductal, and AD cells (acinar cells that have undergone ADM) by flow cytometry, allowing for further study of ADM [5].

YAP1 (Yes-Associated Protein 1) and TAZ (Transcriptional co-activator with a PDZ-binding motif) regulate diverse cellular processes, including proliferation, cell survival, and organ size [6]. Due to the lack of DNA binding activity, YAP1/TAZ need to bind other DNA binding transcription factors to stimulate target gene expression. TEAD (TEA domain family member) family proteins are the major transcription factor partner that can bind YAP1/TAZ [7]. The 3D structural basis of YAP/TEAD interaction was previously reported [8]. YAP1 has been shown to be an important regulator in PDAC progression [9,10,11,12]. YAP1 has been shown to be hyperactivated or overexpressed in pancreatic tumor samples [13,14], and it functions as a critical transcriptional effector downstream of the oncogenic KRAS-MAPK (mitogen-activated protein kinase) pathway to promote neoplastic progression to pancreatic ductal adenocarcinoma in mice [15]. Furthermore, YAP1 activation is an important mechanism in driving pancreatic tumor growth in KRAS-independent PDAC recurrence [14]. However, whether YAP1 is required for ADM and PDAC initiation and progression in human is still unclear. Pancreatic-specific inactivation of the Hippo pathway results in de-differentiation of acinar cells [16,17], suggesting that this pathway may be involved in ADM.

TGFβ1 is a growth factor that plays important roles in mediating chronic inflammation-associated tissue injuries in multiple organs [18,19]. TGFβ1 upregulation also has been well-documented in human pancreatitis [20,21]. The TGFβ signaling pathway has been shown to induce ADM, and SMAD4 is required for ADM in human acinar cells [5]. In addition, The metaplastic AD cells gain transient proliferative capacity that can be extended by KRAS [5], suggesting that the TGFβ pathway promotes PDAC initiation in humans [5,22]. However, the TGFβ signaling pathway plays a completely opposite role in mouse ADM [23,24,25,26], highlighting fundamental differences between mouse and human acinar cells. Thus, it is of great importance to use human tissues for mechanistic studies of the signaling pathways driving human ADM.

In the current study, we tested the hypothesis that the Hippo and TGFβ pathways converge in nuclei to mediate ADM in human cells. Using normal primary human acinar cells, we found that YAP1 and SMAD4 bind to promoters of genes upregulated during ADM. Knocking down of YAP1/TAZ in acinar cells prevents ADM and blocks proliferation of AD cells. These findings suggest that the TGFβ and Hippo signaling pathways cooperate to induce ADM and initiate PDAC.

## 2. Methods

### 2.1. Isolation of Human Pancreatic Cells Using Flow Cytometry

All experiments involving normal human primary pancreatic tissues from organ donors were reviewed by the institutional review board at UT Health San Antonio. The tissues were de-identified, with only information on sex, race, age, weight, height and cause of death. The IRB committee agreed that this project does not require IRB approval because it meets the following criteria: Not human research as defined by DHHS regulations at 45 CFR 46 and FDA regulations at 21 CFR 56; The project does not include non-routine intervention or interaction with a living individual for the primary purpose of obtaining data regarding the effect of the intervention or interaction, nor do the researchers obtain private, identifiable information about living individuals.

Human islet-depleted cell fractions were collected by Prodo Laboratories, Inc, from 32 organ donors (age range 16–60 years, Appendix A) deceased due to acute traumatic or anoxic death. The tissues were shipped overnight to our laboratory and were de-identified, with only information on sex, race, age, weight, height, and cause of death available. Primary pancreatic acinar and ductal cells were isolated from the normal human pancreas as previously described, with some modifications [5]. Islet-depleted human pancreatic exocrine tissue was incubated with FITC-conjugated UEA-1 (0.25 μg/mL) (Vector Laboratories, Newark, CA, USA, FL-1061-5) for 10 min at 4 °C. After washing with PBS, cells were digested with TrypLE™ Express (Life Technologies, Grand Island, NY, USA, 12605-028) for 5–8 min at 37 °C. Cells were collected by centrifugation and washed with FACS buffer (10 mM EGTA, 2% FBS in PBS). After washing, cells were stained with Pacific blue-conjugated anti-CLA (BioLegend, San Diego, CA, USA, 321308) and anti-7AAD (BioLegend, San Diego, CA, USA, 420404) for 15 min at 4 °C. After staining, cells were collected by centrifugation and washed with PBS. Acinar and ductal cells were sorted using a FACSAria™ II (BD Biosciences, San Diego, CA, USA) and collected in 100% FBS. After sorting, cells were washed with serum-free Advanced DMEM/F-12 media (Life Technologies, Grand Island, NY, USA, 12634-010).

### 2.2. Induction of ADM in Sorted Acinar Cells by 2D Culture

To coat plates, 5% Matrigel in advanced DMEM/F-12 (Life Technologies, Grand Island, NY, USA, 12634-010) was placed in wells and incubated at 37 °C for 20 min, then aspirated to leave a thin layer of substrate. Human primary exocrine tissues were plated in the coated well and cultured in Advanced DMEM/F-12 supplemented with 5% FBS. Then, 24 h later, the media was replaced with advanced DMDM/F-12 supplemented with 2% FBS and ROCK inhibitor Y27632 (10 µM, to prevent cell death) for another 5 days. 

### 2.3. Knockdown and Overexpression of YAP1 and TAZ

YAP1 shRNA (CCCAGTTAAATGTTCACCAAT) [27] and TAZ shRNA (CCTGCCGGAGTCTTTCTTTAA) [28] were cloned into pLKO.1, then mCherry or KRAS-mCherry cassette was added. YAP1(5SA) and TAZ(4SA) were constructed into pLenti6-mCherry. Lentiviruses containing these constructs were packaged in 293T cells by co-transfection with packaging plasmids pMD2.G (Addgene, Watertown, MA, USA, #12259) and psPAX2 (Addgene, Watertown, MA, USA, #12260). Infection of cells for knockdown or overexpression of YAP1 and TAZ was performed 24 h after plating cells in 2D format with a supplementation of 8 µg/mL polybrene (Millipore, Burlington, MA, USA, TR1003G) and with a multiplicity of infection of 50. 

### 2.4. Sphere Formation Assay

For sphere formation, the growth factor-reduced Matrigel was mixed with cells suspended in an equal volume of Advanced DMEM/F-12 media. The mixtures (60 μL) were placed around the bottom rim of each well in 48-well plates. After solidification at 37 °C for 20 min, each well was overlaid with 300 μL of Advanced DMEM/F-12 media supplemented with recombinant human (rh) EGF (50 ng/mL, Sigma, St. Louis, MO, USA), rhR-spondin I (500 ng/mL, R&D systems, Minneapolis, MN, USA), rhFGF10 (50 ng/mL, R&D systems), recombinant mouse Noggin (100 ng/mL, R&D systems), and 10 mM Nicotinamide. Media was changed twice per week. The sphere numbers were counted, and fluorescence images were taken using a Lecia DFC365 FX microscope. Images were analyzed with Leica Application Suite X v1.9.0 and Image J v1.46r. 

### 2.5. qRT-PCR Analysis

Total RNA was extracted using the Direct-zol RNA Miniprep Kit (Zymo Research, Irvine, CA, USA) and used for cDNA synthesis with the High Capacity cDNA Reverse Transcription Kit (Applied Biosystems, Waltham, MA, USA) according to the manufacturer’s protocol. Relative mRNA level was measured by qRT-PCR of each cDNA in duplicate with iTaq Universal SYBR Green Supermix (Bio-Rad, Hercules, CA, USA) and the CFX96 Real-Time System (Bio-Rad, Hercules, CA, USA). Normalizations across samples were performed using 18s RNA primers. qPCR Primer sequences are shown in Appendix A. 

### 2.6. Western Blot Analysis

Cells were trypsinized and collected by centrifugation, then washed with PBS and homogenized in RIPA lysis buffer containing phosphatase and protease inhibitors (GenDEPOT, P3100-001 and P3200-001, Baker, TX, USA) on ice. After 10 cycles of sonication (Bioruptor^®^ Pico Sonication System, Cat.B01060001, Denville, NJ, USA), proteins were extracted and the concentration was determined by a BCA kit (Pierce, 23228, ThermoFisher Scientific, Waltham, MA, USA). A total of 50 ug protein per sample was resolved in SDS-PAGE, then transferred to PVDF membrane (ThermoFisher Scientific, Prod # 88520, Waltham, MA, USA) and blocked with 5% milk in PBST (0.1% Tween^®^ 20, Sigma-Aldrich, P1379-500 mL, St. Louis, MO, USA) for one hour at room temperature. Membranes were incubated with primary antibodies (YAP1, Proteintech, 13584-1-AP, Rosemont, IL, USA; TAZ, Proteintech, 23306-1-AP; GAPDH, Santa Cruz Biotechnology, sc-32233, Dallas, TX, USA) at 4° overnight, and then were exposed to secondary antibodies (mouse anti-rabit IgG-HRP, Santa Cruz, SC2357) for one hour. Finally, protein expression was determined by exposing membranes with Clarity™ Western ECL Substrate and capturing luminescence with an Amersham Imager 600 (Cytiva, Marlborough, MA, USA). 

### 2.7. Immunohistochemistry 

Tissue sections were deparaffinized and incubated in Antigen Unmasking Solution (Vector laboratories, H-3300) at 95 °C for 12 min. Sections were treated with methanol containing 3% hydrogen peroxide for 15 min at room temperature to reduce endogenous peroxidase activity and were blocked with blocking buffer containing 10% donkey serum, 1% BSA, and 0.025% Triton X-100 for 45 min at room temperature. The sections were stained with anti-YAP1 (Cell signaling technology, 14074, Danvers, MA, USA) at 4 °C overnight, followed by incubation in Biotinylated Goat anti-rabbit Ig secondary antibody (BD Pharmingen, 550338, Franklin Lakes, NJ, USA) for 60 min at room temperature. Further development was performed with Streptavidin-horseradish peroxidase (BD Pharmingen, 51-75477E) and DAB substrate kit (BD Pharmingen, 550880). Subsequently, the sections were counterstained with hematoxylin for 2 min. 

### 2.8. RNA-Sequencing Sample Preparation

Pellets of sorted cells were collected for total RNA extraction using Zymo Direct-zol RNA Miniprep Kit (Zymo Research, Irvine, CA, USA, R2063) following manufacturer’s instructions. RNA quality was verified by Qubit 4 Fluorometer (Thermo Fisher Scientific, Waltham, MA, USA). Indexed cDNA libraries were prepared using the Illumina^®^ Stranded mRNA Prep Ligation kit (Illumina, San Diego, CA, USA, 20040532) following manufacturer’s instructions. The cDNA libraries were then submitted to the Next Generation Sequencing Core at UT Health San Antonio for high throughput sequencing analysis using an Illumina NovaSeq 6000 System.

### 2.9. ChIP-Seq Sample Preparation

In total, ~1 million cells were cross linked with 1% formaldehyde at room temperature for 10 min, then double cross-linked with 2 mM DSG (ProteoChem, Hurricane, UT, YSA) for 1 h, followed by another 10 min with 1% formaldehyde. Cross-linking was quenched with 0.125 M glycine for 5 min. After cell lysis and nuclear extraction, chromatin was sonicated in lysis buffer with Q800R sonicator (QSonica, Newtown, CT, USA) to acquire ~200–500 bp fragments. Soluble chromatin was incubated with 3–6 mg of antibody at 4 C overnight. The antibodies used in ChIP included anti-YAP1 (14074S, Cell Signaling Technology, Danvers, MA, USA), anti-SMAD4 (38454S, Cell Signaling Technology), and anti-H3K27ac (ab4729, Abcam, Waltham, MA, USA). Immunoprecipitated complexes were pulled down using 50 µL Dynabeads Protein G (Life Technologies, Carlsbad, CA, USA). Beads were washed with washing buffer (50 mM HEPES pH7.6, 1 mM EDTA, 0.7% Na Deoxycholate, 1% NP-40, 0.5 M LiCl) six times to minimize the background. Protein-DNA complexes were then eluted, and DNA fragments were purified. Sequencing was performed using the Illumina’s HiSeq 3000 System. 

### 2.10. ATAC-Seq Sample Preparation

ATAC-seq library preparation was performed on sorted cells (100,000 cells per sample) using Active Motif ATAC-seq kit (Active Motif, Carlsbad, CA, USA, 53150). Cells were incubated with 100 μL ice-cold ATAC-lysis buffer to collect cell nuclei. After centrifugation, nuclei were incubated with the tagmentation master mix in a shaking heat block at 37 °C, 800 rpm for 30 min. The DNA was purified using columns supplied in the kit and amplified for 11 cycles using indexed primers. The PCR products were subjected to size-selection with SPRI bead solution. The DNA libraries were analyzed on an Agilent Technologies 2100-Bioanalyzer (Agilent Technologies, Santa Clara, CA, USA). Quantification of the libraries was performed using the Qubit dsDNA HS Assay Kit (Thermo Fisher Scientific, Waltham, MA, USA, Q32851). Sequencing was performed using the Illumina NovaSeq 6000 System. 

### 2.11. Bioinformatics

RNA sequencing reads were aligned to reference genome GRCh38 using TopHat 2.1.1, and gene expression was quantified using HTSeq 0.11.1. Differential expression analysis of RNA seq data was performed using DeSeq2 1.36.0 package in R software 4.2.0. For two-group comparison, the differentially expressed genes (DEGs) were defined as fold change > 2, and Padj < 0.05. For three-group LRT comparison, the DEGs were defined as Padj < 0.05. K means clustering was performed using the Complexheatmap 2.13.2 package in R in order to identify gene clusters with certain expression patterns. Gene ontology analysis was performed using the clusterProfiler 4.4.4 package in R with default settings. ATAC and ChIP sequencing reads were aligned to reference genome GRCh38 using Bowtie 2.4.4 and. Bigwig files and heatmaps of signal strength were generated using deepTools 3.5.1. Bigwig files were normalized to counts per million (CPM). For peak calling, sequencing files for samples of each lineage were concatenated, and then peak calling was performed using MACS2 2.2.7.1. Motif enrichment analysis and genomic annotation of peaks was performed using Homer 4.11.1. Visualization of signal intensity at individual loci was performed using Integrative Genomics Viewer 2.4.14. 

## 3. Results

### 3.1. Molecular Characterization of Human ADM Using Primary Exocrine Cells

Normal human pancreatic exocrine tissues from organ donors were flow-sorted as previous described [5] to obtain fresh acinar and ductal cells. The sorted acinar cells were then cultured for six days to induce ADM, as previous described [5]. Freshly sorted acinar cells (UEA-1^+^CLA^−^CD133^−^), ductal cells (UEA-1^−^CLA^+^CD133^+^), and cultured AD cells (acinar cells that have undergone ADM) (UEA-1^+^CLA^−^CD133^+^) were subjected to RNA-seq analysis [29], which revealed five gene clusters with distinct expression patterns (Figure 1A, K means cluster = 5). Specifically, cluster 1 genes were downregulated in AD cells compared to fresh acinar cells, including many pancreatic digestive enzymes (*AMY2A*, *CEL*, *CTRC*, *PNLIP*, *PRSS1*, etc.), suggesting the loss of acinar lineage identity of AD cells. However, part of the acinar gene expression pattern was still maintained in short-term induced AD cells (cluster 2, *CXCL17*, *GSTA1*, *RNASE1*, *SPINK1*, etc.; Figure 1B). Cluster 3 included genes that were highly expressed in normal ductal cells but not in acinar cells, and whose expressions were upregulated in AD cells. Many well-established ductal/ADM markers were found here, including *SOX9*, *KRT19*, *PROM1* and *SPP1* (Figure 1C), suggesting that AD cells acquired partial ductal transcriptomic features. Additional genes, such as *CFTR*, *GJA1*, *FSTL1*, and *IGFBP7*, were found in this cluster (Figure 1C), providing more markers for ADM. Cluster 4 included genes highly expressed in AD cells but not in either normal acinar or ductal cells. Moreover, genes highly expressed in ductal cells but not induced in AD cells were identified in cluster 5, including *EGR4*, *HAND1*, *IL1RL1* and *TRIM29* (Figure 1D). While the unique gene expression profile in our AD cells is consistent with previous reports regarding human pancreatic acinar cell dedifferentiation [30,31], this is the first comprehensive transcriptomic comparison of normal acinar, normal ductal, and AD cells, which provided a full spectrum of transcriptomic change during ADM. 

A gene ontology analysis of cluster 1 and cluster 2 revealed enrichment in cytoplasmic translation, small molecule catabolic process, ribosome biogenesis, and assembly programs, corroborating normal pancreatic acinar function (Appendix A). Importantly, gene ontology analysis of cluster 3 of AD-high ductal genes revealed enrichment in the regulation of small GTPase mediated signaling, cell matrix adhesion, and wound healing. The TGFβ receptor signaling pathway and hippo signaling pathway were enriched (Figure 1E), suggesting potential implications of both signaling pathways in human ADM. Genes involved in these pathways that were found in this cluster include *TGFB2*, *SMAD7*, *TGFBR1*, *ITGB6*, *YAP1*, *WWTR1*, *CYR61*, and *CTGF* (Figure 1F,G). 

In addition, genes specifically induced in AD cells were found to be mainly involved in extracellular matrix regulation (cluster 4, Appendix A), while the ductal-specific genes were mainly involved in cell proliferation (cluster 5, Appendix A). Consistent with our findings on the activated Hippo pathway in ductal and AD cells, we stained normal and pancreatitis tissue arrays and found that YAP1 was expressed in ductal cells but not acinar cells in normal human pancreas, as well as in ADM structures in human pancreatitis tissue (Appendix A). This was similar to the staining pattern of SMAD4 that we previous reported [5]. Our data indicate that acinar cells display distinctive downregulation of genes related to digestive function and gain expression of both ductal and AD unique genes during ADM while still maintaining part of their acinar identity. 

### 3.2. YAP1 Binds to Existing Accessible Genomic Regions in Acinar Cells to Induce Expression of ADM Associated Genes

To investigate the contribution of YAP1 as a co-transcriptional modulator, we performed chromatin immunoprecipitation sequencing (ChIP-seq) using an anti-YAP1 antibody on both fresh acinar cells and AD cells after 6 days of culture. To evaluate the epigenetic context of YAP1 binding, ATAC-seq was performed in order to mark regions of open chromatin, and ChIP-seq for histone marker H3K27ac was used to identify regions of transcriptional activity. In the ChIP-seq for YAP1 in AD cells, we detected 23,189 peaks in proximity to 9580 genes, compared to only 295 peaks in proximity to 195 genes in fresh acinar cells. The YAP1 binding at these peaks was substantially stronger in AD cells (Figure 2A). Despite low YAP1 binding, chromatin accessibility and H3K27ac enrichment were observed at these peaks in fresh acinar cells (Figure 2A and Appendix A), implying that the YAP1 binding sites in AD cells were already accessible before ADM. Motif analysis of YAP1 ChIP peaks showed that the most highly enriched transcription factor binding sites were those corresponding to TEADs, the primary transcription factors for the hippo signaling pathway (Figure 2B) [23]. Other enriched motifs included the ATF3/AP-1 complex, which has previously been implicated in ADM [32], and pancreatic development genes, such as GATA6 [33], FOXA1 [34], and RPBJ [35], suggesting that YAP1 acts as a co-regulator of a variety of DNA binding proteins during ADM. Over 80% of YAP1 binding sites were found in introns or distant intergenic regions, suggesting that YAP1 has greater activity at regulatory elements than it does at promoters (Appendix A). Interestingly, almost 50% of AD upregulated genes compared with acinar cells have YAP1 binding sites, while less than 20% of AD downregulated genes have YAP1 binding sites (Figure 2C).

Motif analysis of YAP1 peaks that had a TEAD binding site (10,766 out of the 23,189 peaks) revealed that certain DNA binding proteins, such as BCL6, NFkB, and STAT3, were more associated with YAP1/TEAD binding (Figure 2D). In contrast, ERG, CREB6, and ATF3/AP1 binding sites were more greatly enriched in YAP1 peaks which did not have a TEAD binding site, suggesting that YAP1 might be directly binding to the AP-1 complex and other TFs without TEADs, a phenomena which has been previously observed in other cellular contexts [36,37]. Among the genes induced during ADM, 1053 were bound by both YAP1 and TEADs, while only 436 were solely bound by YAP1, suggesting that YAP1/TEAD interactions may be the main component of Hippo-mediated changes during ADM (Appendix A). Ontology analysis revealed that AD upregulated YAP1 target genes with or without TEADs were enriched for genes related to the extracellular matrix and wound healing. However, YAP1/TEAD interactions specifically regulate ECM receptor interaction, collagen formation, epithelial cell proliferation, and TGFβ signaling pathway, highlighting the distinctive functional effects of YAP1/TEADs transcriptional activity (Appendix A).

Zooming in to specific loci, we found that YAP1 bound to the promoters of genes known to be upregulated in ADM, including *SOX9*, *KRT19*, *FSTL1*, and *CCN2*, where the chromatin was at an open state in acinar, ductal, and AD cells (Figure 2E). Among all YAP1 target genes which were overexpressed in AD cells compared to fresh acinar cells, approximately half were also highly expressed in ductal cells, while half were exclusive to AD cells (Figure 2F). Gene ontology of these 2 gene sets revealed that both were enriched for genes related to cell adhesion, wound healing, and extracellular matrix organization (Figure 2G). Genes which were expressed in ductal cells and upregulated in AD cells were enriched for genes related to ERK signaling, integrin signaling, and epithelial morphology, while genes which were only upregulated in AD cells were enriched for genes related to WNT signaling, glycolysis, and protein glycosylation (Figure 2G). Taken together, these results show that, while YAP1 mediated transcriptional changes during ADM partially result in upregulation of ductal cell genes in stressed acinar cells, likely facilitating the adoption of ductal-like morphology, there are also ADM-specific transcriptional programs which are activated via YAP1. 

### 3.3. YAP1 Co-Activation with SMAD4 Induces ADM Associated Transcriptional Changes

RNA-seq analysis showed that the ductal specific genes induced in AD cells were enriched for Hippo and TGFβ pathways (Figure 1B), suggesting a potential interplay between these two signaling pathways in human ADM. In line with this, one of the enriched motifs in YAP1 binding sites in AD cells was that of the transcription factor SMAD4 (Figure 2B). To access the role of SMAD4 in ADM associated gene expression changes, we conducted ChIP-seq using an anti-SMAD4 antibody on AD cells. YAP1 binding was notably strong at SMAD4 peaks (Figure 3A). These SMAD4 peaks exhibited open chromatin and H3K27ac enrichment even in fresh acinar cells (Appendix A). Among the 5581 SMAD4 binding peaks, the plurality (38%) were located in gene promoter regions (Appendix A). SMAD4-bound genes were more frequently upregulated or downregulated in AD cells regardless of whether SMAD4 bound to the promoter (Appendix A). Motif analysis showed that SMAD4 peaks contained many of the same enriched features as YAP1 peaks, including ATF3/AP-1, TEADs, GATA6, and NRF2 (Figure 3B). Out of the 3763 genes in proximity to a SMAD4 peak, 2840 were also bound by YAP1 (Figure 3C). Co-binding of YAP1 and SMAD4 was observed in the promoters of ADM-associated genes, including *ID1*, *HES1*, and *WNT7A* and *B* (Figure 3D). Among the genes co-bound by YAP1/SMAD4, 538 were upregulated in AD cells compared to fresh acinar cells, while 161 were downregulated (Figure 3E). This implies that SMAD4 upregulates ADM associated genes both directly as a transcription factor and through interaction with regulatory genetic elements. These findings suggest that YAP1 and SMAD4 collaboratively regulate transcriptional changes during ADM, with SMAD4 primarily acting as a transcriptional activator.

Gene ontology analysis revealed that genes induced during ADM and co-bound by SMAD4 and YAP1 were primarily associated with cell adhesion, wound healing, epithelial cell proliferation, TGFβ signaling, and response to growth factors (Figure 3F). Another set of genes that were notably enriched among ADM-induced YAP1/SMAD4 targets included stress and regeneration associated cytokines including *FGF2* and members of the *TNF*, and *WNT* families (Figure 3G). These data suggest that YAP1 and SMAD4 coordinate to induce transcriptional upregulation of genes related to stress response and proliferative healing.

### 3.4. YAP1/TAZ Cooperated with TGFβ Signaling to Promote Acinar Proliferation during Acinar to Ductal Metaplasia 

To study the role of YAP1/TAZ in primary human acinar cells, we generated YAP1^5SA^/TAZ^4SA^ lentiviral vectors featuring constitutively active YAP1 and TAZ mutants [28] that are resistant to inhibition by the upstream Hippo pathway, as well as an mCherry marker (Appendix A). We sorted fresh acinar cells and infected them in 2D culture using either a control mCherry lentivirus or the YAP^5SA^/TAZ^4SA^-mCherry lentivirus (Appendix A). Because AD cells can secrete TGFβ [5], we blocked the endogenous TGFβ signaling with the inhibitor SB431542 (Figure 4A). Under each infection condition, cells were treated with either a control vehicle or 1 μM SB431542. The YAP1 and TAZ mRNA levels were significantly elevated 72 h post-infection with YAP1^5SA^ and TAZ^4SA^ lentiviruses (Appendix A). The YAP1/TAZ targets, *CTGF/CCN2*, and *CYR61*/*CCN1* also showed upregulation, suggesting successful infection of human acinar cells with YAP1^5SA^ and TAZ^4SA^ (Appendix A). Treatment with SB431542 resulted in a slight but statistically insignificant decrease in YAP1/TAZ expression (Appendix A). The expression of *CYR61* showed a similar trend to YAP1/TAZ. The expression of *CTGF* showed a significant reduction post-SB431542 treatment, aligning with previous findings that *CTGF* is also downstream of TGFβ signaling [38]. 

We previously showed that acinar cells undergoing ADM acquire transient proliferative capacity [5]. To investigate whether active YAP1/TAZ can promote acinar cell proliferation, we performed a sphere formation assay as previously described [5]. Consistent with our earlier findings that acinar cells have limited proliferative capacity [5], cells infected with control vectors barely formed spheres, regardless of whether they were treated with SB431542, on both Day 7 and Day 14 (Figure 4B). In contrast, acinar cells infected with YAP^5SA^/TAZ^4SA^ formed ring-like spheres that continued to grow for two weeks, indicating that YAP1/TAZ significantly enhanced their proliferation (Figure 4B,C). Interestingly, acinar cells infected with YAP^5SA^/TAZ^4SA^ and treated with SB431542 formed fewer and smaller spheres compared to the YAP1/TAZ overexpression group treated with vehicle. Most of these spheres diminished between Day 7 to Day 14 (Figure 4B,C). This suggests that, while YAP1/TAZ can induce acinar cell proliferation, this effect can be attenuated by blocking endogenous TGFβ signaling.

### 3.5. YAP1/TAZ Are Required in TGFβ-Induced Acinar to Ductal Metaplasia 

Previously, we found that TGFβ can induce ADM in human acinar cells. To further investigate the role of YAP1/TAZ in TGFβ-induced ADM, we constructed a lentiviral vector containing shYAP-shTAZ along with mCherry as a marker (Appendix A). To knockdown YAP1 and TAZ in acinar cells, we sorted fresh acinar cells and infected them in 2D culture with either control mCherry lentivirus or the shYAP-shTAZ-mCherry lentivirus (Appendix A). Under each infection condition, cells were also treated with either a control vehicle or 1 nM TGFβ to induce ADM (Figure 5A). 0As in the overexpression assay, we observed a greater than 70% infection efficacy in each experimental group, as indicated by the mCherry expression (Appendix A). qRT-PCR analysis showed reduced YAP1/TAZ mRNA levels following a 72 h infection with shYAP1/shTAZ lentivirus. This led to the downregulation of YAP1/TAZ target genes *CTGF* and *CYR61* (Appendix A). Treatment with TGFβ slightly, but not significantly, increased the expression levels of YAP1/TAZ and their targets. 

In the sphere formation assay, we found that acinar cells infected with either scramble shRNA or shYAP-shTAZ lentivirus barely formed spheres (Figure 5B,C). Cells infected with scramble shRNA and treated with TGFβ did produce some small spheres on Day 7, suggesting TGFβ-induced ADM. However, most of these spheres died off by Day 14, consistent with our previous observations. In contrast, knocking down YAP1/TAZ in TGFβ-treated acinar cells resulted in no spheres being formed either on Day 7 or Day 14 (Figure 5B,C). Taken together, our data suggest that YAP1/TAZ is required for TGFβ-induced ADM.

### 3.6. YAP1/TAZ Are Required in Maintaining ADM State and the KRAS-Driven Proliferation in AD Cells

We next investigated whether YAP1/TAZ are essential for maintaining AD cell state and for KRAS-induced proliferation in AD cells. We added shYAP-shTAZ to a previous oncoKRAS-mCherry lentiviral vector [5], allowing us to knock down YAP1/TAZ and overexpress oncogenic KRAS and mCherry simultaneously (Appendix A). We then cultured fresh tissue in 2D culture condition for 6 days to induce ADM, followed by sorting the AD cells for viral infection (Figure 6A). The cells were collected after 5 days of infection for qRT-PCR and sphere formation assay. Unlike acinar cells, the 2D-cultured AD cells exhibited a long and spindle shaped morphology, similar to the morphological features of ductal cells (Appendix A). We observed an infection efficiency greater than 80%, which was higher than that in acinar cells. After infection with KRAS-mCherry lentivirus, there was an upregulation of YAP1/TAZ and their targets, suggesting that KRAS may induce YAP1/TAZ expression. However, the expression of YAP1/TAZ and their targets decreased when infected with shYAP-shTAZ-KRAS-mCherry virus (Appendix A). 

Consistent with previous sphere formation results [5], AD cells formed small spheres by 7 days, and most of these spheres began to die off after 10 days in culture, suggesting the transient activation of proliferation in AD cells. However, AD cells with YAP1/TAZ knockdown barely formed spheres on either Day 7 or on Day 14, indicating that YAP1/TAZ are required for AD cell proliferation (Figure 6B,C). Furthermore, AD cells transduced with KRAS-mCherry virus gained prolonged proliferation, forming larger and more spheres. In contrast, knocking down YAP1/TAZ led to smaller and fewer spheres even in the presence of oncogenic KRAS, with most of them dying off by Day 14 (Figure 6B,C). Taken together, our data suggest that YAP1/TAZ are required to maintain the transient proliferation capacity in AD cells and are essential for KRAS-induced extended proliferation of AD cells. 

## 4. Discussion

Acinar-to-Ductal Metaplasia (ADM) is often triggered by pancreatic inflammation, such as pancreatitis, and serves as a cellular response for tissue repair [2]. Key signaling pathways, including TGF-β [5,22,39] and Hippo [17,40], are activated or inhibited under stressful conditions. This alters the change of gene expression profiles of acinar cells, resulting in morphological changes where the cells lose their specialized acinar features and adopt characteristics of ductal cells. The surrounding cellular environment, consisting of the extracellular matrix and other cell types such as stellate and immune cells, also plays a role in promoting the ADM process [41]. Ultimately, these metaplastic cells may proliferate to form duct-like structures. For the first time, we have been able to examine the intricate molecular alterations in primary human exocrine tissues during ADM. While this is an in vitro setting and not an in vivo one, it represents the closest approximation to human physiology that has been achieved to date.

By isolating acinar, ductal, and AD cells, we can delve deeper into the molecular landscape of ADM in human pancreatic exocrine tissues. Our study reveals a complex transition between acinar and AD states, characterized by five distinct gene clusters. Notably, genes in cluster 1 related to digestive functions are downregulated in AD cells while some acinar-specific genes of cluster 2 are retained, suggesting an intermediate or plastic state. This downregulation may serve as a protective mechanism to mitigate the harmful effects of digestive enzymes on tissues and the body during pancreatic stress [42]. It raises questions about how these cells lose their acinar lineage identity and what controls downregulation of acinar specific genes. Furthermore, the group of genes highly expressed in normal ductal cells and upregulated in AD cells is particularly interesting, providing the rationale for the term ADM itself. *SOX9* and *KRT19* are the most well-known marker genes for identifying ADM [43]. We have now discovered an expanded array of ductal genes that are expressed during ADM. Genes in cluster 4, unique to AD cells, are primarily involved in extracellular matrix regulation. These changes could be partially attributed to stress induced by culture conditions as induced genes may vary based on the type of stress. Genes in cluster 5 are specific to ductal cells, suggesting that AD cells have not fully transitioned into ductal cells. This finding could significantly impact our understanding of the origins of PDAC. Our findings strongly suggest that ADM represents an intermediate state with its own distinct transcriptomic signature. 

The cooperation between the TGFβ pathway and the Hippo signaling pathway is a complex and multifaceted interaction that plays vital roles in various cellular processes, including regulation of cell growth, differentiation, and apoptosis. The cooperation between the two pathways can occur at multiple levels [44]. The TGF-β pathway can influence the Hippo pathway effectors YAP1/TAZ by facilitating their nuclear localization [45]. YAP1/TAZ also affects SMAD cellular localization [46], and SMAD potentiates the signaling events of the TGFβ/SMAD pathway [47]. However, how these two pathways converge in the human pancreas is elusive. In our present study, we embarked on a detailed investigation which revealed that the Hippo and TGF-β pathways converge within nuclei to mediate Acinar-to-Ductal Metaplasia in human cells. We previously showed that phopho-SMAD2/3 co-stain with acinar marker Amylase (*AMY2*) in human pancreatitis tissue [5], suggesting the activation of the TGFβ pathway in ADM. Additionally, both our research and that of others have found that YAP1 is expressed only in ductal cells and not in acinar cells in a normal human pancreas. However, YAP1 is expressed in ADM structures in human pancreatitis tissue [40], supporting the idea that the two pathways converge during ADM. 

Utilizing a sphere formation assay as an experimental model, we discovered that the overexpression of constitutively activated YAP1/TAZ can promote ADM. However, this effect is contingent upon the presence of TGFβ signaling. To further validate this finding, we knocked down YAP1/TAZ in acinar cells and observed that ADM was effectively halted, even when exogenous TGFβ was provided. This finding underscores the crucial role that YAP1/TAZ plays in the initiation and progression of ADM. When we extended this knockdown experiment to AD cells, the sphere formation was completely inhibited. This offers strong evidence that YAP1/TAZ proteins are required for AD cell proliferation. Knocking down YAP1/TAZ also inhibits the growth of AD cells induced by oncogenic KRAS, consistent with the observation in mouse PDAC models [9,15]. Our data strongly support the notion that YAP1/TAZ and TGFβ signaling pathway interplay during ADM.

In our genome-wide analyses of YAP1 binding sites in AD cells using ChIP-seq, we found that most YAP1 binding regions align with enhancer elements, consistent with previous studies [37,48,49]. The analysis highlights the interplay between YAP1 and transcription factors such as TEADs, ATF3/AP-1, GATA6, and FOXA1, indicating a complex regulatory mechanism. Our study also showed YAP1 binding in ADM-induced ductal genes like *SOX9*, *KRT19* and *PROM1*. Interestingly, the chromatin of these genes is in an open state in acinar cells, in which these genes are not expressed, suggesting that ADM-induced genes are in an open state and poised for expression during ADM. This allows for immediate transcriptional activation of YAP1 targets upon nuclear entry under stress conditions, consistent with previous observations that certain YAP1 binding partners, such as ATF3, can initiate transcriptional changes within hours of the onset of pancreatitis [32].

The YAP1 ChIP peaks contributing to ADM-associated transcriptional changes were almost evenly split between those that contained at least one TEAD binding motif and those that did not. Motif analysis showed that the binding sites of some ADM-associated transcription factors, such as the ATF3/AP-1 complex, were more enriched in YAP1 peaks lacking a TEAD binding site compared to the peaks that had one, suggesting that YAP1 may directly bind to the ATF3/AP-1 complex and other TFs without TEADs, as has been previously reported [36]. However, YAP1/TEAD/AP-1 complexes are also common [37]. Thus, YAP1 contributes to ADM transcriptional changes through both TEAD dependent and independent mechanisms, and the general function of these two gene sets is noticeable different. Our study also revealed that YAP1 is directly associated with critical genes related to cell adhesion, wound healing, and extracellular matrix organization, underlying its vital role in cellular morphology and survival [50]. 

We have shown previously that the ADM in human acinar cells is induced by TGFβ [5]. Both we and others have found SMAD4 and YAP1 nuclei staining in ADM of human pancreatitis tissue [5,40]. The RNA-seq analysis and subsequent ChIP-seq investigations depict the collaboration between YAP1 and SMAD4 in controlling gene expression. A large proportion of the genes bound by SMAD4 also have binding sites for YAP1, further affirming this partnership. The most enriched binding site in SMAD4 ChIP peaks was ATF3, suggesting that, like YAP1, SMAD4 cooperates in the rapid initiation of ADM upon stress in acinar cells. This is in line with previous demonstrations that TGF-β rapidly induces ADM, given that SMAD4 transduces TGF-β receptor signaling. The genes influenced by YAP1/SMAD4 co-regulation were enriched in pathways associated with cell adhesion, wound healing, epithelial cell proliferation, and stress response. The relationship between YAP1 and SMAD4 in inducing ADM-associated transcriptional changes adds a new dimension to our understanding of Hippo and TGFβ pathways in human ADM.

TGF-β is well-known for its role in wound healing and tissue repair and has been shown to induce fibrosis in various organs [51]. Similarly, the Hippo pathway, particularly through its effectors YAP1/TAZ, has been implicated in the regulation of fibrosis [17]. Our findings indicate that the cooperative interaction between these pathways can influence the balance between regeneration and fibrosis in injured pancreas. Moreover, we identified a group of cytokine genes upregulated in AD cells. Interestingly, half of them are YAP1 targets, while the other half have both YAP1 and SMAD4 binding sites. This suggests that YAP1 takes a central role, but also that the coordinated action between TGF-β and Hippo pathways may play a crucial part in regulating immune responses, including both immune tolerance and inflammation.

Considering that obtaining human pancreatic ADM tissues is difficult, the in vitro ADM model we presented in this study using normal human primary pancreatic cells offers an alternative approach to decipher the mechanism of human ADM. However, the question remains as to how well the in vitro model can recapitulate the process in patients. It would be interesting to compare our in vitro samples with clinical ADM samples to validate the clinical relevance.

## 5. Conclusions

In conclusion, this study offers new perspectives on the complex interplay between the Hippo and TGF-β pathways in the context of ADM. By elucidating how YAP1/TAZ and SMAD4 interaction regulates gene expression during ADM, we reveal the molecular mechanisms through which these two pathways control acinar cells to undergo ADM and drive PDAC initiation. These insights not only expand our knowledge of these essential biological processes but also present promising avenues for future research and potential clinical applications. Our research further raises questions about the downstream effects of this regulation on other signaling pathways, and how these contribute to ADM and PDAC progression also warrant further exploration.

## Figures and Tables

**Figure 1 cells-13-00186-f001:**
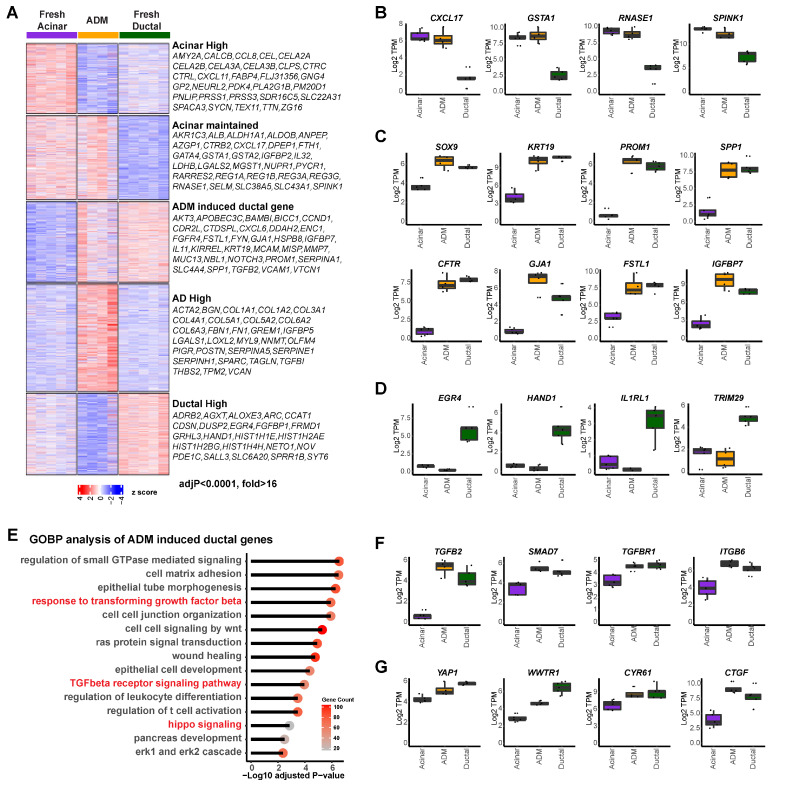
Unique transcription profile of human primary acinar cells during ADM. (**A**) Heatmap of gene expression in acinar cells (n = 5), ductal cells (n = 5), and AD cells (n = 4), clustered by expression pattern (K means = 5) among sample groups. Right: highly differentially expressed genes found in each of the five clusters (adjusted *p* value < 0.0001, fold change > 16). (**B**) Expression of indicated genes which were highly expressed in acinar cells and maintained during ADM. (**C**) Expression of indicated genes which were highly expressed in ductal cells and induced in AD cells. (**D**) Expression of indicated genes which were highly expressed in ductal cells but not in acinar or AD cells. (**E**) Gene ontology analysis of genes highly expressed in ductal cells and induced in AD cells. (**F**) Expression of TGFβ signaling associated genes in acinar, AD, and ductal cells. (**G**) Expression of Hippo signaling associated genes in acinar, AD, and ductal cells.

**Figure 2 cells-13-00186-f002:**
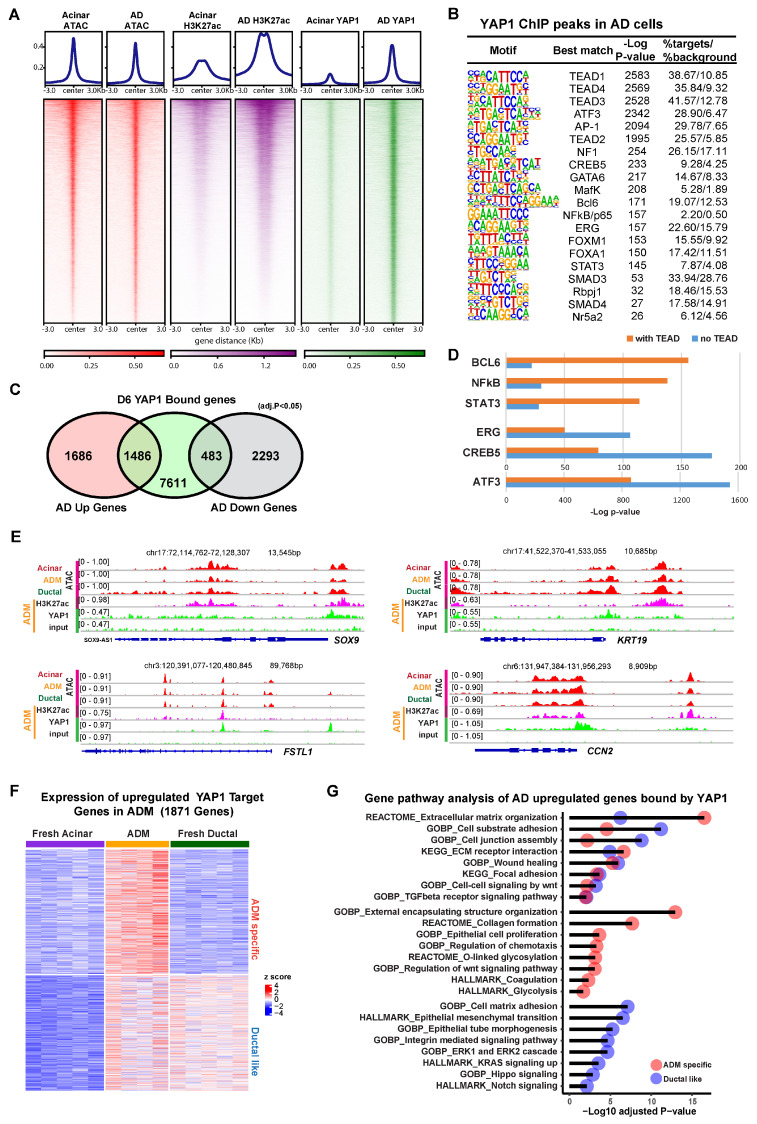
YAP1 binds to accessible chromatin regions to induce ADM-associated transcriptional programs. (**A**) Heatmap showing ATAC-seq signal, H3K27ac binding, and YAP1 binding at YAP1 ChIP peaks in fresh acinar cells and AD cells. (**B**) Results of motif analysis of YAP1 ChIP peaks in AD cells. (**C**) Venn diagram showing overlap between YAP1 bound genes in AD cells and genes which are upregulated or downregulated (adjusted *p* value < 0.05, fold change > 2) in AD cells relative to fresh acinar cells. (**D**) Enrichment *p*-values for indicated motifs in YAP1 peaks with or without TEAD binding motif. (**E**) IGV tracks at the loci of indicated ADM associated genes, showing ATAC signal (of fresh acinar cells, AD cells, and ductal cells), as well as H3K27ac and YAP1 ChIP signals (in AD cells). (**F**) Expression of YAP1 bound genes that are upregulated (adjusted *p* value < 0.05, fold change > 2) in AD cells compared to fresh acinar cells. The genes were clustered as 2 groups based on their expression pattern (ADM specific, and Ductal like). (**G**) Results of gene ontology analysis of genes described in (**F**).

**Figure 3 cells-13-00186-f003:**
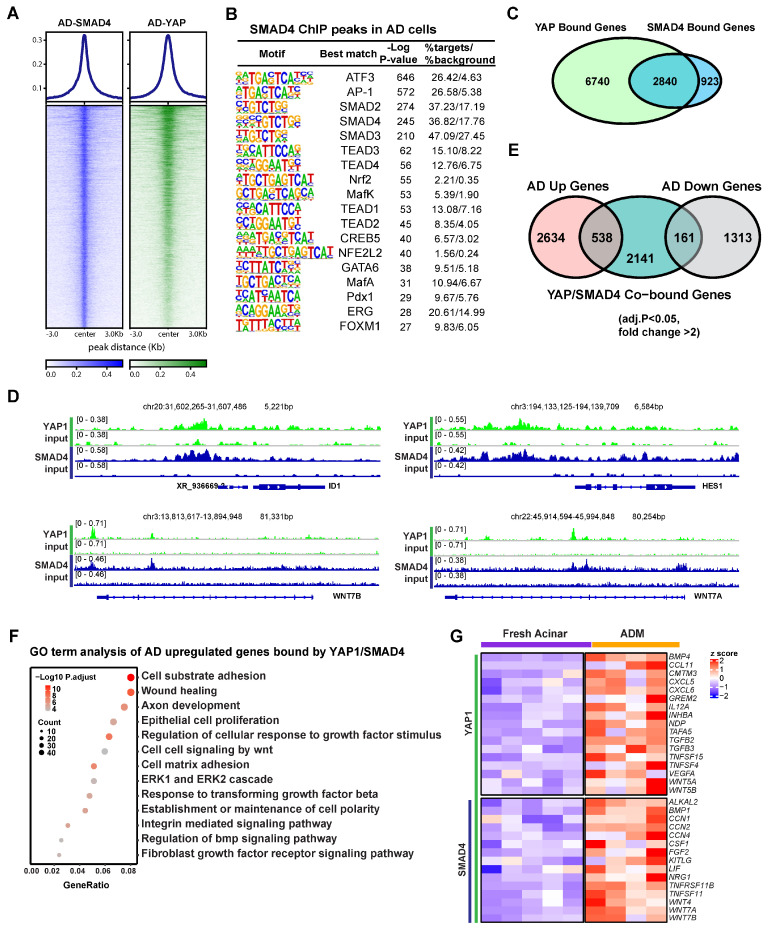
SMAD4 and YAP cooperate to induce ADM associated transcriptional changes. (**A**) Heatmap showing binding of SMAD4 and YAP1 at SMAD4 ChIP peaks in AD cells. (**B**) Motif analysis of SMAD4 ChIP Peaks in AD cells. (**C**) Venn diagram showing overlap of YAP1 bound genes and SMAD4 bound genes in AD cells. (**D**) IGV tracks showing YAP1 and SMAD4 binding at the loci of ADM associated genes. (**E**) Venn diagram showing overlap between AD induced or repressed genes and genes co-bound by YAP1 and SMAD4. (**F**) Gene ontology of genes which are upregulated in AD cells and co-bound by YAP1 and SMAD4. (**G**) Expression heatmap of ADM-induced, cytokine-associated genes in AD cells and fresh acinar cells. Gene symbols in red are co-bound by YAP1 and SMAD4.

**Figure 4 cells-13-00186-f004:**
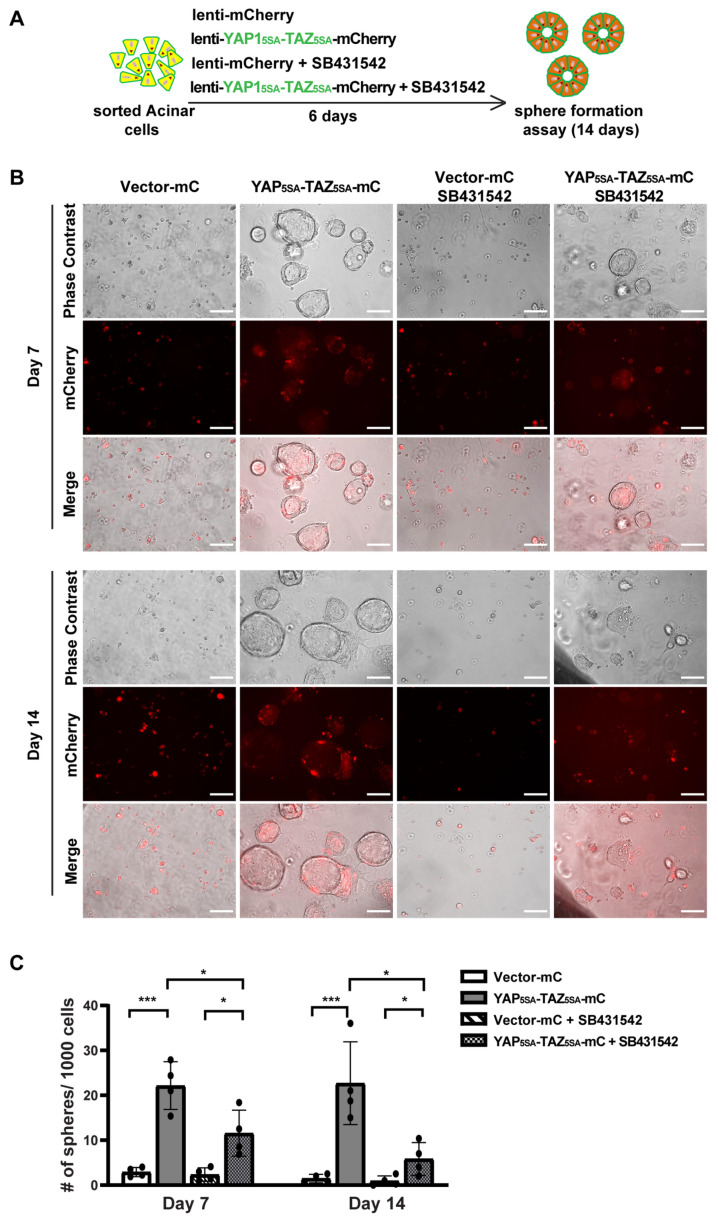
YAP1/TAZ overexpression induces acinar cell proliferation, and TGFβ inhibitor blocks YAP1/TAZ-induced proliferation. (**A**) Schematic of experiment strategies. (**B**) Representative images of sphere formation of acinar cells transfected with vector-mCherry or YAP1/TAZ-mCherry, and treated with or without SB431542, on Day 7 and Day 14. Scale bar, 100 µm. (**C**) Quantification results of sphere formation on Day 7 and Day 14. N = 4–6. Error bars = S.E.M. * *p* < 0.05, *** *p* < 0.001.

**Figure 5 cells-13-00186-f005:**
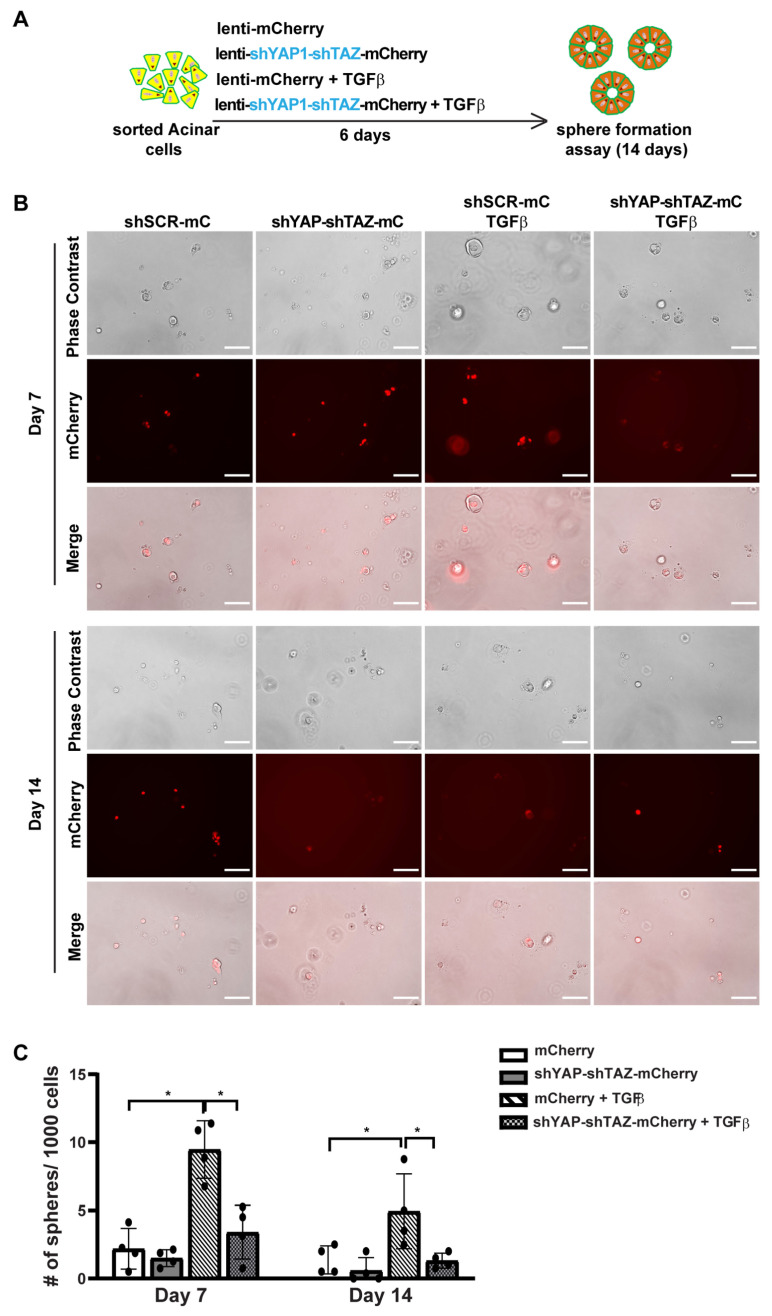
YAP1/TAZ knockdown suppresses ADM-induced sphere formation. (**A**) Schematic illustration of experimental design. (**B**) Representative images of sphere formation of acinar cells transfected with shScramble-mCherry or shYAP1/shTAZ-mCherry, and treated with or without TGFβ, on Day 7 and Day 14. Scale bar, 100 µm. (**C**) Quantification of sphere formation on Day 7 and Day 14. N = 4–6. Error bars = S.E.M. * *p* < 0.05.

**Figure 6 cells-13-00186-f006:**
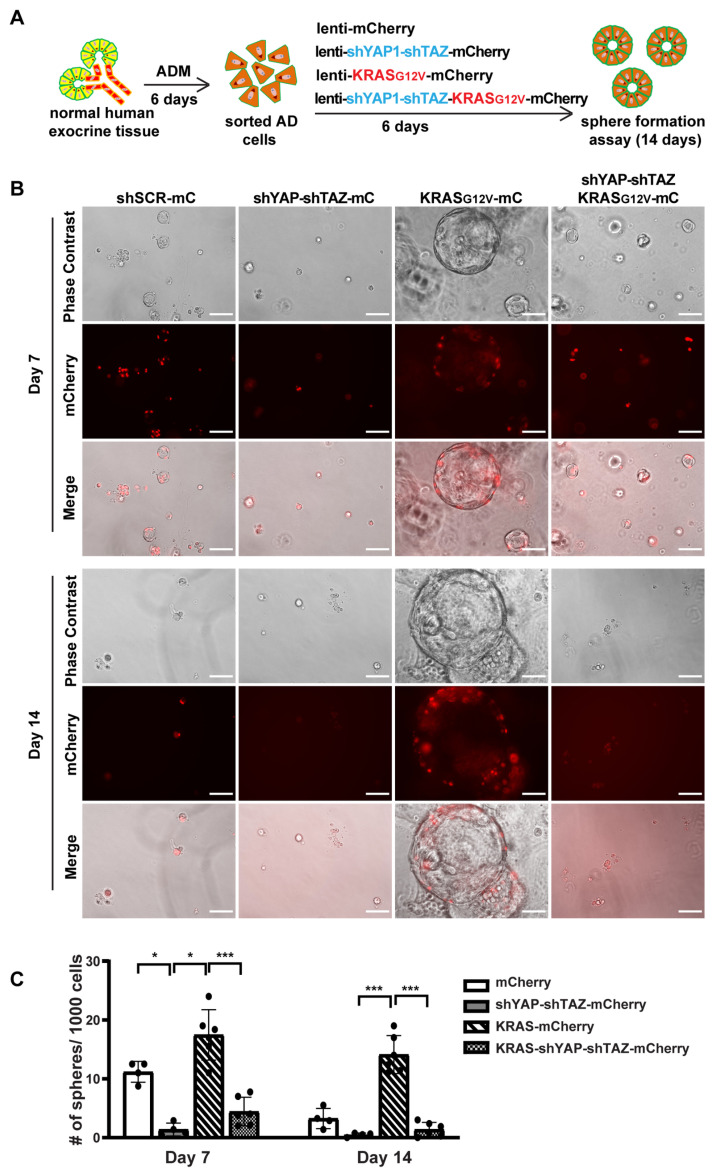
YAP1/TAZ knockdown suppresses oncogenic KRAS induced sphere formation. (**A**) Schematic illustration of experimental design. (**B**) Representative images of AD cell sphere formation under different virus infection conditions on Day 7 and Day 14. N = 4–6. Scale bar, 100 µm. (**C**) Quantification results of acinar cell sphere formation on Day 7 and Day 14. N = 4–6. Error bars = S.E.M. * *p* < 0.05, *** *p* < 0.001.

## Data Availability

The raw and processed high throughput sequencing data described in this study are deposited at Gene Expression Omnibus (accession numbers GSE222991), which will be made publicly available as of the date of publication.

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
