# Peer review of "TGFβ and Hippo Signaling Pathways Coordinate to Promote Acinar to Ductal Metaplasia in Human Pancreas"

_cells, 2024, doi:10.3390/cells13020186_

Round 1

Reviewer 1 Report

Comments and Suggestions for Authors

Authors have addressed role of YAP1 and SMAD4 as markers of HIPPO and TGFb signaling in possible association with Acinar to Ductal metaplasia in pancreatitis. With involvement of YAP1 and SMAD4 ADM may progress to PDAC. Study conducted is extensive and well designed. However, minor but important issues may be addressed as below.

1. YAP-TEAD complex may be well explained in introduction section, as study revolves around that theme and experiments are aimed to look at association of TEAD. 

Relevant references (Li et al, 2010 Genes and Development) may be cited. doi: 10.1101/gad.1865810

2. Ethics concern: Normal human pancreas tissues were used in the study. Ethics committee approval needs to be mentioned in methods as well as separate section above acknowledgement.

3. How many individual enrolled in study for collection of normal tissues, need to be mentioned too.

4. ADM cells may be checked if bear tumorigenic potential in presence of absence of TGFb inhibitor, using animal model to validate the hypothesis. This experiments if already done can be included as can be justified if cannot be done as part of this study.

Author Response

  1. YAP-TEAD complex may be well explained in introduction section, as study revolves around that theme and experiments are aimed to look at association of TEAD. Relevant references (Li et al, 2010 Genes and Development) may be cited. doi: 10.1101/gad.1865810

Response: Thanks for this thoughtful suggestion. Indeed, we reported our findings regarding YAP-TEAD interaction based on our Chip-seq and ATAC-seq experiments, and we agree this should be explained in the Introduction section. Therefore, we revised the Introduction section to briefly describe YAP-TEAD complex, and cited the reference suggested by the reviewer.

  1. Ethics concern: Normal human pancreas tissues were used in the study. Ethics committee approval needs to be mentioned in methods as well as separate section above acknowledgement.

Response: Thanks for the comment. The normal human primary pancreatic tissues of organ donors used in the experiments were reviewed by the institutional review board at UT Health San Antonio. The tissues were de-identified, with only information on sex, race, age, weight, height and cause of death. The IRB committee has agreed that this project does not require IRB approval because it is either: Not human research as defined by DHHS regulations at 45 CFR 46 and FDA regulations at 21 CFR 56; The project does not include non-routine intervention or interaction with a living individual for the primary purpose of obtaining data regarding the effect of the intervention or interaction, nor do the researchers obtain private, identifiable information about living individuals.

We now include this explanation in the Methods as well as in a separate section above acknowledgement.

  1. How many individual enrolled in study for collection of normal tissues, need to be mentioned too.

Response: Total of 32 donor tissues were used throughout the study. We now include a supplementary data file to provide the population characteristics of each individual donor (sex, age, race, etc).

  1. ADM cells may be checked if bear tumorigenic potential in presence of absence of TGFb inhibitor, using animal model to validate the hypothesis. This experiments if already done can be included as can be justified if cannot be done as part of this study.

Response: Thanks for the suggestion. The tissues we received are from normal organ donor. We did not detect KRAS mutation from our RNA-seq data. In fact, in our other projects, we further engineered the ADM cells to simultaneously introduce four common PDAC oncogenic mutations (oncogenic KRAS, loss of function of p16, p53 and SMAD4), and confirmed the tumorigenic potential of these mutated cells in xenograft mice model. These results have been submitted to Nature Communications and was just accepted for publication in principle. Therefore, we believe that ADM, while by itself may not be sufficient to induce transformation, may represent the first step of tumorigenesis with oncogenic stimulus. Unfortunately, as those results are still in editorial editing process in Nature Communications, we are not able to cite or discuss them in the present manuscript.

Reviewer 2 Report

Comments and Suggestions for Authors

This is a very interesting and timely study of the interactions between transforming growth factor beta (TGF-b) and Hippo pathways in promoting acinar to ductal metaplasia (ADM) in acinar cells derived from the normal human pancreas, the roles of YAP1/TAZ and SMAD4 in this process, the intersection with KRAS pathways, and differences between acinar and ductal cells. The study consists of a series of experiments that rely on sensitive cell isolation techniques of acinar and ductal cells, conversion of acinar cells to ADM cells (termed AD cells), genomic analysis and ChIP-Seq studies, cell culture studies, and silencing as well as overactivation studies. While the findings point to the need for a great deal of additional mechanistic work that should be carried out for more definitive interpretations of the current studies, I believe that the current findings will be of great interest to the researchers in this field and the scientific community and will stimulate new lines of investigation. As such, the paper represents an important advance.

Minor recommendations:

The authors should indicate that Y27632 is a ROCK inhibitor (line 97), and they should indicate why they used this inhibitor to induce ADM in vitro.

The authors should provide clearer histograms figures since it was difficult to delineate the low-value bars without reading the legends very carefully.

The authors should mention that a limitation of their current findings is the use AD cells that were generate in vitro. They should also provide the age and sex of the normal pancreas donors, how the tissues were obtained, and whether regulatory approval was obtained or not required.

Author Response

  1. The authors should indicate that Y27632 is a ROCK inhibitor (line 97), and they should indicate why they used this inhibitor to induce ADM in vitro..

Response: Thanks for the suggestion. In fact, the ROCK inhibitor is supplemented in the growth medium to prevent cell death of primary human cells, especially after flow sorting stress. It is not for inducing ADM. We revised the corresponding Methods section to clarify this.

  1. The authors should provide clearer histograms figures since it was difficult to delineate the low-value bars without reading the legends very carefully..

Response: Thanks for the comment. High resolution figures for Figure 2 and Figure 3 are now provided.

  1. The authors should mention that a limitation of their current findings is the use AD cells that were generate in vitro. They should also provide the age and sex of the normal pancreas donors, how the tissues were obtained, and whether regulatory approval was obtained or not required

Response: We appreciate the reviewer’s concern regarding the limitation of our in vitro model. Considering obtaining the human pancreatic ADM tissues is of great difficulty, the in vitro ADM model we presented in this study using normal human primary pancreatic cells offers an alternative approach to decipher the mechanism of human ADM. However, we acknowledge that the question remains as to how well the in vitro model can recapitulate the process in patients. It would be interesting to compare our in vitro samples with clinical ADM samples to validate the clinical relevance. We revised the Discussion section to briefly mention this limitation.

We revised the Method section to include the requested details regarding human organ donor information. The population characteristics (age, sex, etc) for the organ donors are provided as a supplementary data file.